# Spinor Field in FLRW Cosmology

**Bijan Saha** [1,2] 

1    Laboratory of Information Technologies, Joint Institute for Nuclear Research, Dubna, 141980 Moscow, Russia; bijan@jinr.ru
2    Institute of Physical Research and Technology, Peoples' Friendship University of Russia (RUDN University), 6 Miklukho-Maklaya Street, 117198 Moscow, Russia

**Abstract:** Within the scope of a Friedmann-Lemaitre-Robertson-Walker (FLRW) cosmological model we study the role of a nonlinear spinor field in the evolution of the universe. In doing so, we exploit the FLRW models given in both Cartesian and spherical coordinates. It is found that if the FLRW model is given in the spherical coordinates the energy-momentum tensor (EMT) of the spinor field possesses nontrivial non-diagonal components, which is not the case for Cartesian coordinates. These non-diagonal components do not depend on either the spinor field nonlinearity or the parameter $k$ that defines the type of curvature of the FLRW model. The presence of such components imposes some restrictions on the spinor field. The problem is studied for open, flat and close geometries and the spinor field is used to simulate different types of sources including dark energies. Some qualitative numerical solutions are given.

**Keywords:** Alexander Friedmann; expanding universe; accelerated expansion; dark energy; spinor field; energy-momentum tensor

## 1. Introduction

The isotropy of cosmic microwave background (CMB) radiation, first detected by the Cosmic Background Explorer (COBE) satellite [1], and further supported by the Wilkinson Microwave Anisotropy Probe (WMAP) data [2], together with the assumption that we are not in any special position in Universe, underlines the Cosmological Principle. According to this principle we live in a homogeneous and isotropic Universe which mean all the space-time points of our Universe can be treated as the center of the Universe and all the directions are equal. Such a Universe is given by a FLRW model. The present day experimental data suggest that our Universe is indeed isotropic one and homogeneous in large scale. That is why the study of present day Universe is dominated by the FLRW model. Exact solution to the Einstein equation found by Russian mathematician A.A. Friedmann suggested that our Universe is expanding. He also observed that there can be three types of solutions: closed, flat and open [3,4]. But those days physicists believed that the Universe is static and unchanging. So Einstein dully rejected Friedmann solutions and introduced cosmological constant into the system to secure a steady solution to his equation. Recall that before Einstein the Universe was thought to be geocentric or heliocentric, which possesses center. But it was Einstein who first told that there is no specific point and any point of the space-time can be the center of the Universe, thus bringing revolutionary changes about the idea of space-time. Even he failed to accept the concept of a Universe that is changing with time. In 1929 Edwin Hubble experimentally showed that the Universe is expanding and there are many galaxies outside our milky way [5]. It buried the idea of a static Universe. Further this model was independently developed by Lemaitre [6], Robertson [7–9] and Walker [10]. So this model is also known as FLRW model. The FLRW model has not only mathematical simplicity, but also experimental support.

Thanks to its ability to simulate different kinds of matter such as perfect fluid, dark energy etc. spinor field is being used by many authors not only to describe the late time

acceleration of the expansion, but also to study the evolution of the Universe at different stages [11–17]. It was found that the spinor field is very sensitive to spacetime geometry. Depending on the concrete type of metric the spinor field may possess different types of nontrivial non-diagonal components of the energy-momentum tensor. As a result the spinor field imposes various kinds of restrictions on both the spacetime geometry and the spinor field itself [18]. Recently spinor field is used in astrophysics to see whether its specific behavior can shed any new light in the study of the objects like black hole and wormhole. Such studies were carried out within the scope of spherically symmetric [19,20] and cylindrically symmetric spacetime [21,22].

Since the present-day universe is surprisingly isotropic and the presence of nontrivial non-diagonal components of the spinor field leads to the severe restrictions on the spinor field, we have studied role of a spinor field in Friedmann-Lemaitre-Robertson-Walker (FLRW) model as well. But in those cases the space-time was given in Cartesian coordinates. In order to see influence of the coordinate transformations on spinor field some works were done by us earlier [23,24]. In this paper we will compare the results founded for FLRW model given in Cartesian and spherical coordinates and study the behavior of the spinor field under such coordinate transformations.

## 2. Basic Equation

Let us consider the action of the gravitational and nonlinear spinor field in the form

$$S = \int \sqrt{-g} \left[ \frac{R}{2\kappa} + L_{sp} \right] d\Omega, \tag{1}$$

where $\kappa = 8\pi G$ is Einstein's gravitational constant, $R$ is the scalar curvature. The spinor field Lagrangian $L_{sp}$ is given by [25]

$$L_{sp} = \frac{\imath}{2} \left[ \bar{\psi} \gamma^\mu \nabla_\mu \psi - \nabla_\mu \bar{\psi} \gamma^\mu \psi \right] - m \bar{\psi} \psi - \lambda F(K). \tag{2}$$

Here, the nonlinear term $F(K)$ is constructed as some arbitrary functions of invariants generated from the real bilinear forms, where $K$ takes one of the following expressions $\{I, J, I + J, I - J\}$. Here $I = S^2$ and $J = P^2$ are the invariants of bilinear spinor forms with $S = \bar{\psi} \psi$ and $P = \imath \bar{\psi} \bar{\gamma}^5 \psi$ being the scalar and pseudo-scalar, respectively. In (2) $\lambda$ is the self-coupling constant. The covariant derivatives of spinor field takes the form [25]

$$\nabla_\mu \psi = \partial_\mu \psi - \Omega_\mu \psi, \quad \nabla_\mu \bar{\psi} = \partial_\mu \bar{\psi} + \bar{\psi} \Omega_\mu, \tag{3}$$

with $\Omega_\mu$ being the spinor affine connections defined by [25]

$$\Omega_\mu = \frac{1}{4} \left( \bar{\gamma}_a \gamma^\beta \partial_\mu e_\beta^{(a)} - \gamma_\rho \gamma^\beta \Gamma^\rho_{\mu\beta} \right). \tag{4}$$

In (4), $\Gamma^\beta_{\mu\alpha}$ is the Christoffel symbol and the Dirac matrices in curve and flat space–time $\gamma$ and $\bar{\gamma}$ are connected to each other in the following way

$$\gamma_\beta = e_\beta^{(b)} \bar{\gamma}_b, \quad \gamma^\alpha = e_{(a)}^\alpha \bar{\gamma}^a. \tag{5}$$

Here, the tetrad vectors $e_\beta^{(b)}$ are related to the metric in the following way

$$g_{\mu\nu}(x) = e_\mu^{(a)}(x) e_\nu^{(b)}(x) \eta_{ab}, \tag{6}$$

and $e_{(a)}^\alpha$ are the inverse to $e_\mu^{(a)}(x)$:

$$e^{\alpha}_{(a)}e^{(a)}_{\beta} = \delta^{\alpha}_{\beta}, \quad e^{\alpha}_{(a)}e^{(b)}_{\alpha} = \delta^b_a. \tag{7}$$

Here, $\eta_{ab} = \text{diag}(1, -1, -1, -1)$ is the Minkowski spacetime. The $\gamma$ matrices obey the following anti-commutation rules

$$\gamma_{\mu}\gamma_{\nu} + \gamma_{\nu}\gamma_{\mu} = 2g_{\mu\nu}, \quad \gamma^{\mu}\gamma^{\nu} + \gamma^{\nu}\gamma^{\mu} = 2g^{\mu\nu}. \tag{8}$$

Varying the Lagrangian (2) with respect to $\bar{\psi}$ and $\psi$, respectively, we obtain the following spinor field equations

$$\imath\gamma^{\mu}\nabla_{\mu}\psi - m\psi - \lambda\mathcal{D}\psi - \imath\lambda\mathcal{G}\bar{\gamma}^5\psi = 0, \tag{9}$$

$$\imath\nabla_{\mu}\bar{\psi}\gamma^{\mu} + m\bar{\psi} + \lambda\mathcal{D}\bar{\psi} + \imath\lambda\mathcal{G}\bar{\psi}\bar{\gamma}^5 = 0, \tag{10}$$

where $\mathcal{D} = 2F_K S, \quad \mathcal{G} = 2F_K P$. It can be shown that in view of the spinor field equations (9) and (10) the spinor field Lagrangian (2) can be expressed as

$$L = \lambda(2KF_K - F), \quad F_K = dF/dK.$$

In this report, we consider the spinor field that depends only on time, i.e., $\psi = \psi(t)$. In view of (3), the energy momentum tensor of the spinor field is defined in the following way [25].

$$T_{\mu}^{\rho} = \frac{\imath}{4}g^{\rho\nu}\left(\bar{\psi}\gamma_{\mu}\partial_{\nu}\psi + \bar{\psi}\gamma_{\nu}\partial_{\mu}\psi - \partial_{\mu}\bar{\psi}\gamma_{\nu}\psi - \partial_{\nu}\bar{\psi}\gamma_{\mu}\psi\right)$$

$$- \frac{\imath}{4}g^{\rho\nu}\bar{\psi}\left(\gamma_{\mu}\Omega_{\nu} + \Omega_{\nu}\gamma_{\mu} + \gamma_{\nu}\Omega_{\mu} + \Omega_{\mu}\gamma_{\nu}\right)\psi - \delta^{\rho}_{\mu}L. \tag{11}$$

It should be noted that the non-diagonal components of the EMT arises thanks to the second term in (11).

The gravitational field is given by isotropic and homogeneous cosmological model proposed by Friedmann, Lemaitre, Robertson and Walker. We consider two cases when the model is given in Cartesian and spherical coordinates. We do it to show that the spinor field is even sensible to the coordinate transformations. Variation of the action (1) with respect to $g^{\mu\nu}$ leads to Einstein equation

$$G^{\nu}_{\mu} = -\kappa T^{\nu}_{\mu}. \tag{12}$$

In what follows, we consider the homogeneous and isotropic cosmological gravitational field given by FLRW model.

**Case I** Let us first consider the FLRW model given in Cartesian coordinates:

$$ds^2 = dt^2 - a^2(t)\left[dx^2 + dy^2 + dz^2\right], \tag{13}$$

where the scale factor $a(t)$ is a function of time only. This case was thoroughly studied in [18,26].

In view of (6) we choose the tetrad in the form

$$e^{(0)}_0 = 1, \quad e^{(1)}_1 = a(t), \quad e^{(2)}_2 = a(t), \quad e^{(3)}_3 = a(t).$$

Then, from (4) we find the following expressions for spinor affine connection

$$\Omega_0 = 0, \quad \Omega_1 = \frac{\dot{a}}{2}\bar{\gamma}^1\bar{\gamma}^0, \quad \Omega_2 = \frac{\dot{a}}{2}\bar{\gamma}^2\bar{\gamma}^0, \quad \Omega_3 = \frac{\dot{a}}{2}\bar{\gamma}^3\bar{\gamma}^0. \tag{14}$$

Thanks to the fact that, in this case $\Omega_1 = \Omega_2 = \Omega_3$ the EMT of the spinor field possesses only diagonal components with [18]:

$$T_0^0 = m_{\rm sp}S + \lambda F(K), \quad T_1^1 = T_2^2 = T_3^3 = \lambda(F(K) - 2KF_K). \tag{15}$$

The absence of non-diagonal components of the EMT leads to the fact that the spinor field does not impose any kind of restriction either on the space-time geometry or on the spinor field. The spinor field equation in this case takes the form

$$\imath\bar{\gamma}^0\left(\dot{\psi} + \frac{3}{2}\frac{\dot{a}}{a}\psi\right) - m_{\rm sp}\psi - \lambda\mathcal{D}\psi - \imath\lambda\mathcal{G}\bar{\gamma}^5\psi = 0, \tag{16}$$

$$\imath\left(\dot{\bar{\psi}} + \frac{3}{2}\frac{\dot{a}}{a}\bar{\psi}\right)\bar{\gamma}^0 + m_{\rm sp}\bar{\psi} + \lambda\mathcal{D}\bar{\psi} + \imath\lambda\mathcal{G}\bar{\psi}\bar{\gamma}^5 = 0. \tag{17}$$

The foregoing system was solved exactly and given in explicit form in [18]. The Einstein field Equation (12) in this case coincide with those considered in the **case II** for $k = 0$. The Einstein equation was solved for different types on nonlinearity.

**Case II** Let us now consider the case when the FLRW model is given in spherical coordinates [27]:

$$ds^2 = dt^2 - a^2(t)\left[\frac{dr^2}{1-kr^2} + r^2d\vartheta^2 + r^2\sin^2\vartheta d\varphi^2\right], \tag{18}$$

with $k$ taking the values $+1$, $0$ and $-1$ which corresponds to a close, flat and open universe, respectively. The purpose of doing this is to show that the spinor field is not only sensitive to space-time geometry, given by different metrics, but also to coordinate transformations. In view of (6), we choose the tetrad in the form

$$e_0^{(0)} = 1, \quad e_1^{(1)} = \frac{a}{\sqrt{1-kr^2}}, \quad e_2^{(2)} = ar, \quad e_3^{(3)} = ar\sin\vartheta.$$

Then, from (5) we find the following $\gamma$ matrices

$$\gamma^0 = \bar{\gamma}^0, \quad \gamma^1 = \frac{\sqrt{1-kr^2}}{a}\bar{\gamma}^1, \quad \gamma^2 = \frac{\bar{\gamma}^2}{ar}, \quad \gamma^3 = \frac{\bar{\gamma}^3}{ar\sin\vartheta}.$$

Further from $\gamma_\mu = g_{\mu\nu}\gamma^\nu$ one finds the $\gamma_\mu$ as well. From (4) in this case we find the following expressions for spinor affine connection

$$\Omega_0 = 0, \tag{19}$$

$$\Omega_1 = \frac{1}{2\sqrt{1-kr^2}}\dot{a}\bar{\gamma}^1\bar{\gamma}^0, \tag{20}$$

$$\Omega_2 = \frac{1}{2}r\dot{a}\bar{\gamma}^2\bar{\gamma}^0 + \frac{1}{2}\sqrt{1-kr^2}\bar{\gamma}^2\bar{\gamma}^1, \tag{21}$$

$$\Omega_3 = \frac{1}{2}\dot{a}r\sin\vartheta\bar{\gamma}^3\bar{\gamma}^0 + \frac{1}{2}\sqrt{1-kr^2}\sin\vartheta\bar{\gamma}^3\bar{\gamma}^1 + \frac{1}{2}\cos\vartheta\bar{\gamma}^3\bar{\gamma}^2. \tag{22}$$

In view of (19)–(22), the spinor field equations can be written as

$$\dot{\psi} + \frac{3}{2}\frac{\dot{a}}{a}\psi + \frac{\sqrt{1-kr^2}}{ar}\bar{\gamma}^0\bar{\gamma}^1\psi + \frac{\cot\vartheta}{2ar}\bar{\gamma}^0\bar{\gamma}^2\psi + \imath(m + \lambda\mathcal{D})\bar{\gamma}^0\psi + \lambda\mathcal{G}\bar{\gamma}^5\bar{\gamma}^0\psi = 0, \tag{23}$$

$$\dot{\bar{\psi}} + \frac{3}{2}\frac{\dot{a}}{a}\bar{\psi} - \frac{\sqrt{1-kr^2}}{ar}\bar{\psi}\bar{\gamma}^0\bar{\gamma}^1 - \frac{\cot\vartheta}{2ar}\bar{\psi}\bar{\gamma}^0\bar{\gamma}^2 - \imath(m + \lambda\mathcal{D})\bar{\psi}\bar{\gamma}^0 + \lambda\mathcal{G}\bar{\psi}\bar{\gamma}^5\bar{\gamma}^0 = 0, \tag{24}$$

The solution to the spinor field equation can be given in the form [18]

$$\varphi(t) = T\exp\left(-\int_t^{t_1} A_1 d\tau\right), \tag{25}$$

where we introduce $\varphi = a^{3/2}\psi$. In the foregoing expression $T = \varphi(t_1)$ is the solution at $t = t_1$. In case of a nonzero spinor mass one can assume $\varphi(t_1) = \mathrm{col}\big(e^{-\imath m t_1}, e^{-\imath m t_1}, e^{\imath m t_1}, e^{\imath m t_1}\big)$, whereas for a massless spinor field $\varphi(t_1) = \mathrm{col}\big(\varphi_1^0, \varphi_2^0, \varphi_3^0, \varphi_4^0\big)$ with $\varphi_i^0$ being constants. In (25) the matrix $A_1 \equiv A$ with $m = 0$ or $\mathcal{D}_1 = \mathcal{D}$, where

$$A = \begin{pmatrix} -\imath\mathcal{D}_1 & 0 & -\lambda\mathcal{G} & B_1 \\ 0 & -\imath\mathcal{D}_1 & B_1^* & -\lambda\mathcal{G} \\ \lambda\mathcal{G} & B_1 & \imath\mathcal{D}_1 & 0 \\ B_1^* & \lambda\mathcal{G} & 0 & \imath\mathcal{D}_1 \end{pmatrix} \tag{26}$$

with $\mathcal{D}_1 = (m + \lambda\mathcal{D})$, $B_1 = -\frac{\sqrt{1-kr^2}}{ar} + \imath\frac{\cot\vartheta}{2ar}$ and $B_1^* = -\frac{\sqrt{1-kr^2}}{ar} - \imath\frac{\cot\vartheta}{2ar}$. It can be shown that $\det A = \big(\mathcal{D}_1^2 + \lambda^2\mathcal{G}^2 - B_1 B_1^*\big)^2$. We can choose the nonlinearity in such a way that the corresponding determinant becomes nontrivial.

In this case from (11) we find the following non-trivial components of the energy momentum tensor of the spinor field

$$T_0^0 = mS + \lambda F, \tag{27}$$

$$T_1^1 = T_2^2 = T_3^3 = -\lambda(2KF_K - F), \tag{28}$$

$$T_3^1 = \frac{a\cos\vartheta}{4\sqrt{1-kr^2}}A^0, \tag{29}$$

$$T_1^0 = \frac{\cot\vartheta}{4r\sqrt{1-kr^2}}A^3, \tag{30}$$

$$T_2^0 = -\frac{3}{4}\sqrt{1-kr^2}\,A^3, \tag{31}$$

$$T_3^0 = \frac{3}{4}\sqrt{1-kr^2}\,\sin\vartheta A^2 - \frac{1}{2}\cos\vartheta A^1. \tag{32}$$

From (27)–(32), we conclude that the diagonal components of the EMT are the same as in previous case. Moreover, in this case the energy-momentum tensor of the spinor field contains nontrivial non-diagonal components. The non-diagonal components

- do not depend on the spinor field nonlinearity;
- occur due to the spinor affine connections;
- appear depending on space-time geometry as well as the system of coordinates;
- impose restrictions on spinor field and/or space-time geometry;
- depend on the value of $k$ which defines the type of curvature, though do not vanish ever for $k = 0$.

It should be emphasized that for a FRW model given in Cartesian coordinates the EMT have only diagonal components with all the non-diagonal one being identically zero [26]. So in this case the non-diagonal components arise as a result of coordinate transformation. Let us also note that all the cosmological space-time given by diagonal metrics such as Bianchi type $VI$, $VI_0$, $V$, $III$, $I$ $LRS - BI$ and $FRW$, possess the same diagonal components of EMT, while possess nontrivial non-diagonal elements who differ from each other for different cases [18]. Moreover non-diagonal metrics such as Bianchi type $II$, $VIII$ and $IX$ also have nontrivial non-diagonal components of EMT. Hence we see that the appearance of the non-diagonal components of the energy-momentum tensor takes place either due to coordinate transformations or space-time geometry.

The components of the EMT of the spinor field contains some spinor field invariants. To define those invariants we write the system of equations for the invariants of the spinor field

$$\dot{S}_0 + 2\mathcal{G}A_0^0 = 0, \tag{33}$$

$$\dot{P}_0 - 2(m + \mathcal{D})A_0^0 = 0, \tag{34}$$

$$\dot{A}_0^0 + 2\mathcal{G}S_0 + 2(m + \mathcal{D})P_0 + 2\frac{\sqrt{1 - kr^2}}{ar}A_0^1 + \frac{\cot\vartheta}{ar}A_0^2 = 0, \tag{35}$$

$$\dot{A}_0^1 + 2\frac{\sqrt{1 - kr^2}}{ar}A_0^0 = 0, \tag{36}$$

$$\dot{A}_0^2 + \frac{\cot\vartheta}{ar}A_0^0 = 0, \tag{37}$$

that gives the following relation between the invariants:

$$P_0^2 - S_0^2 + \left(A_0^0\right)^2 - \left(A_0^1\right)^2 - \left(A_0^2\right)^2 = C_0, \quad C_0 = \text{Const.} \tag{38}$$

In (33)–(38) the quantities with a subscript "0" are related to the normal ones as follows: $X_0 = Xa^3$. From (38) we can conclude that since $C_0$ is an arbitrary constant, the each term of (38) should be constant as well.

Let us recall that the Einstein tensor $G_\mu^\nu$ corresponding to the metric (18) possesses only nontrivial diagonal components. Hence from (12) we obtain the following non-diagonal expressions

$$0 = T_\mu^\nu, \quad \mu \neq \nu. \tag{39}$$

In view of (29)–(32) from (39), one dully finds that

$$A^0 = 0, \quad A^3 = 0, \quad A^1 = (3/2)\sqrt{1 - kr^2}\tan\vartheta A^2. \tag{40}$$

It is worth noting that, if the FRW model given by the Cartesian coordinates the non-diagonal components of EMT are identically zero, hence relation such as (40) does not exist.

We are now ready to consider the diagonal components of the Einstein system of equations which for the metric (18) takes the form

$$2\frac{\ddot{a}}{a} + \left(\frac{\dot{a}^2}{a^2} + \frac{k}{a^2}\right) = 8\pi G T_1^1, \tag{41}$$

$$3\left(\frac{\dot{a}^2}{a^2} + \frac{k}{a^2}\right) = 8\pi G T_0^0. \tag{42}$$

The system (41) and (42) coincides the corresponding system for the FLRW metric given by cartesian coordinates in case of $k = 0$. One can solve (42) to find $a$, but to take into account both equations (42) and (41) it is better to combine them and rewrite (41). In view of (27) and (28) then we obtain

$$\ddot{a} = -\frac{\kappa}{6}(mS - 2\lambda F + 6\lambda K F_K)a. \tag{43}$$

The equation (43) does not contain $k$ that defines the type of space-time curvature, hence it is true for both cases. But in order to take this very important quantity $k$ into account we have to exploit (42) as the initial condition for $\dot{a}$:

$$\dot{a} = \pm\sqrt{(\kappa/3)(mS + \lambda F)a^2 - k}, \tag{44}$$

Now, we can solve (43) with the initial condition given by (44). It comes out that these equations are consistent if one takes sign "−" in (44). Alternatively, one can solve (44), but for the system to be consistent he has to check whether the result satisfies (43). Exploiting (33)–(37) it was shown that [18,26]

$$K = \frac{V_0^2}{a^6}, \quad V_0 = \text{const.}, \tag{45}$$

which is true for $K = \{J, \, I+J, \, I-J\}$ for a massless spinor field, whereas, for $K = I$ it is valid both for massless and massive spinor field. Thus, $S$, $K$, hence $F(K)$ are the functions of $a$. Hence given the spinor field nonlinearity the foregoing equation can be solved either analytically or numerically. The first integral of (43) takes the form

$$\dot{a} = \sqrt{\int f(a)da + C_c}, \tag{46}$$

where we define $f(a) = -\frac{\kappa}{3}(mS - 2\lambda F + 6\lambda K F_K)a$ and $C_c$ is a constant which should be defined from (44). The solution to the equation (46) can be given in quadrature

$$\int \frac{da}{\sqrt{\int f(a)da + C_c}} = t. \tag{47}$$

In what follows we solve the system (41) and (42) numerically and in doing so we rewrite the system in the following way

$$\dot{a} = Ha, \tag{48}$$

$$\dot{H} = -\frac{3}{2}H^2 - \frac{1}{2}\frac{k}{a^2} - \frac{\kappa}{2}\lambda(2KF_K - F), \tag{49}$$

$$H^2 = \frac{\kappa}{3}(mS + \lambda F) - \frac{k}{a^2}, \tag{50}$$

where $H$ is the Hubble constant. As one sees, in the foregoing system the first two are differential equations, whereas the third one is a constraint, which we use as the initial condition for $H$:

$$H = \pm\sqrt{\kappa(mS + \lambda F)/3 - k/a^2}. \tag{51}$$

Since the expression under the root must be non-negative, it imposes some restrictions on the choice of the initial value of $a$ as well.

## 3. Numerical Solutions

In what follows we solve the system (48)–(50) numerically. In doing so, we consider several cases nonlinearity of the spinor field, that describes various types of sources such as perfect fluid and dark energy.

### 3.1. Barotropic Equation of State

It should be noted that prior to 1998, when the late time accelerated mode of expansion of the Universe was detected, perfect fluid was the most popular form of matter used to study the evolution of the Universe. But after 1998 cosmologists first considered $\Lambda$-term to explain the new phenomenon, then in analogy with perfect fluid they proposed quintessence which can be implemented by the barotropic equation of state (EoS). This equation gives a linear dependence between the pressure and energy density and was exploited by many authors [28–31]. The spinor description of perfect fluid, quintessence, $\Lambda$-term, phantom matter etc. were simulated by the nonlinear term [18,26]

$$F(S) = \lambda S^{1+W} - m_{\text{sp}}S, \quad \lambda = \text{const.}, \tag{52}$$

in the spinor field Lagrangian (2). Depending on the value of $W$, the Equation (52) can give rise to both perfect fluid, such as dust, radiation etc. and dark energy such as quintessence, cosmological term, phantom matter etc. For $W \in [0,1]$, it describes a perfect fluid. The

value $W = -1$ represents a typical cosmological constant ($\Lambda$-term) [32–34], whereas $W \in [-1, -1/3]$ gives rise to a quintessence, while for $W < -1$ it ascribes a phantom matter.

Let us now solve (48)–(50) numerically for the nonlinear term given by (52). We consider both massive and massless spinor field. The values of $W$ are taken to be $1/3$, $-1/2$ and $-1$ describing the radiation, quintessence and cosmological constant, respectively. For simplicity we set $S_0 = 1$, $G = 1$, $\lambda = 0.5$ here and in the cases to follow. We also set $m_{\rm sp} = 0$ for a massless and $m_{\rm sp} = 1$ for a massive spinor field.

In Figure 1 we have illustrated the evolution of the Universe filled with radiation, given by a massless spinor field. In the figures the blue solid line stands for a closed universe given by $k = 1$, red dash-dot line stands for a flat universe with $k = 0$ and black long dash line stands for an open universe with $k = -1$.

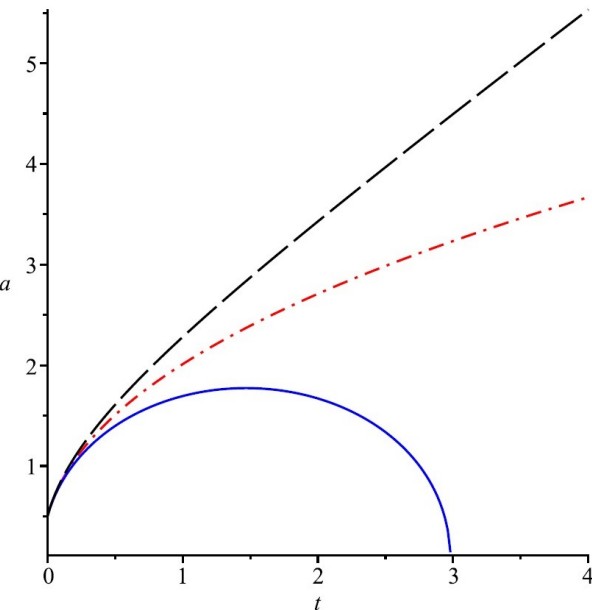

**Figure 1.** Evolution of the FRW Universe (scale factor $a$) in presence of a radiation given by a massless spinor field. Here solid blue, dash-dot red and long dash black lines correspond to $k = 1, 0, -1$, respectively.

We have also considered the case with the spinor field nonlinearity describing a quintessence ($W = -1/2$) and cosmological constant ($W = -1$). Both massive and massless spinor fields are taken into account. Since in both cases the energy density is less than the critical density, independent to the value of $k$ we have only open type of universe. The behavior of the evolution is qualitatively same as that of in case of a modified Chaplygin gas. The corresponding figures will be similar to those in Figure 2, only the rate of expansion being much slower.

### 3.2. Chaplygin Gas

In order to combine two different physical concepts such as dark matter and dark energy, and thus reduce the two physical parameters in one, a rather exotic equation of state was proposed in [35] which was further generalized in the works [36,37]. It was shown that such kind of dark energy can be modeled by the massless spinor field with the nonlinearity [18]

$$F = \left(A + \lambda S^{1+\alpha}\right)^{1/(1+\alpha)}, \tag{53}$$

where $A$ is a positive constant and $0 < \alpha \leq 1$.

We have solved (48)–(50) numerically for the nonlinear term given by (53). We consider only massless spinor field setting $m_{\text{sp}} = 0$. The parameters $S_0$, $G$ and $\lambda$ were taken as in previous case. We have also set $A = 1/2$ and $\alpha = 1/3$.

As in case of quintessence and cosmological constant, the evolution of the universe filled with Chaplygin gas is qualitatively same as in case of a modified Chaplygin gas which are illustrated in Figure 2. The expansion rate in this case is higher than the previous case but slower than in the case to follow.

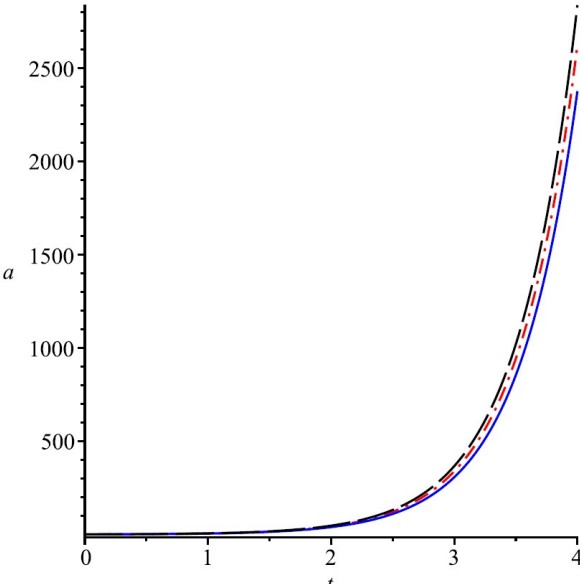

**Figure 2.** Evolution of the FRW Universe (scale factor *a*) in presence of a modified Chaplygin gas given by a massless spinor field. As one sees, due to the presence of dark energy for all values of *k* we have open universe. Here solid blue, dash-dot red and long dash black lines correspond to $k = 1, 0, -1$, respectively.

*3.3. Modified Chaplygin Gas*

Though the dark energy and the dark matter act in a completely different way, many researchers suppose that they are different manifestations of a single entity. Following such an idea a modified Chaplygin gas was introduced in [38] and was further developed in [39]. The modified Chaplygin gas can be generated by a massless spinor field with the nonlinearity given by [18]

$$F = \left[ \frac{A}{1 + W} + \lambda S^{(1+\alpha)(1+W)} \right]^{1/(1+\alpha)}. \tag{54}$$

with $W$ being a constant, $A > 0$ and $0 \le \alpha \le 1$. In fact, mathematically it is a combination of quintessence and Chaplygin gas. We have solved (48)–(50) numerically for the nonlinear term given by (54). Since we consider only massless spinor field, we set $m_{\text{sp}} = 0$. For simplicity we set $S_0$, $G$, $\lambda$, $A$, and $\alpha$ as in previous cases. Beside that we set $W = -1/2$.

In Figure 3 we have illustrated the evolution of the universe when the universe is filled with nonlinear spinor field simulating a modified Chaplygin gas.

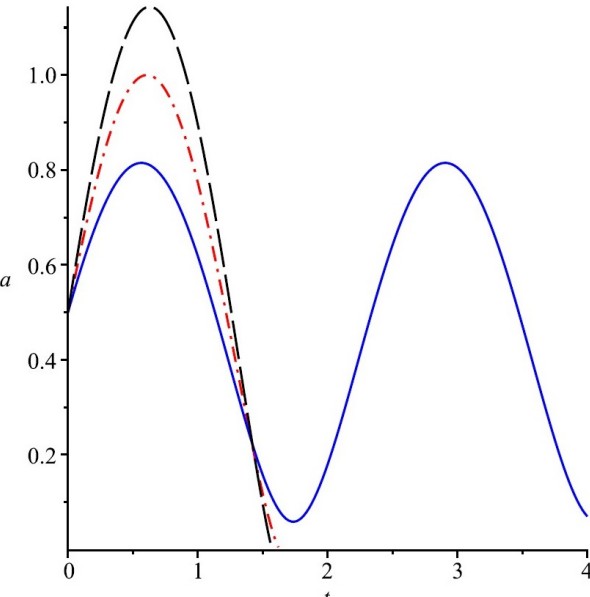

**Figure 3.** Evolution of the FRW Universe (scale factor *a*) in presence of a modified quintessence given by a massless spinor field. In this case the value of *k* plays definite role. Here solid blue, dash-dot red and long dash black lines correspond to $k = 1, 0, -1$, respectively.

*3.4. Modified Quintessence*

A modified Quintessence was proposed in order to avoid eternal acceleration of the universe. In some cases it gives cyclic universe that pops up from a Big Bang singularity, expands to some maximum value and then decreases and finally ends in Big Crunch. In some cases it might be periodic without singularity. A spinor description of a modified quintessence was proposed in [31]

$$p = W(\varepsilon - \varepsilon_{\mathrm{cr}}), \quad W \in (-1, 0), \tag{55}$$

with $\varepsilon_{\mathrm{cr}}$ being some critical energy density. The model gives rise to cyclic or oscillatory universe. Setting $\varepsilon_{\mathrm{cr}} = 0$ one obtains ordinary quintessence. As one sees from (55), the pressure is negative as long as $\varepsilon > \varepsilon_{\mathrm{cr}}$. Since with the expansion of the universe the energy density decreases, at some moment of time $\varepsilon$ becomes less than $\varepsilon_{\mathrm{cr}}$, i.e., $\varepsilon < \varepsilon_{\mathrm{cr}}$. This leads to the positive pressure and the contraction of the universe. It can be shown that a modified quintessence can be modeled by a spinor field nonlinearity

$$F = \lambda S^{1+W} + \frac{W}{1+W}\varepsilon_{\mathrm{cr}}. \tag{56}$$

In this case while solving the system (48)–(50) we consider values of the parameters as in case of quintessence. For critical density we set $\varepsilon_{\mathrm{cr}} = 1$.

In Figure 3 we have illustrated the evolution of the universe when the universe is filled with nonlinear massless spinor field simulating a modified quintessence. It should be emphasized that in this case both massless and massive the spinor field can be considered.

**4. Conclusions and Discussions**

Within the scope of a FLRW cosmological model we have studied the role of a nonlinear spinor field in the evolution of the universe. It is found that if the FLRW model is given in spherical coordinates the spinor field possesses nontrivial non-diagonal components of the EMT, whereas is case of Cartesian coordinates these components are trivial. Since the Einstein tensor in this case is diagonal, the presence of nontrivial non-diagonal components of the EMT imposes some restrictions on the components of spinor field. Corresponding

equations are solved and the results are graphically illustrated for the cases when the universe is filled with radiation, modified Chaplygin gas and modified quintessence.

As it was already noticed, the coordinate transformation from Cartesian to spherical coordinates gives rise to non-diagonal components of EMT that owe to spinor affine connections. This very fact suggests that the definition of spinor affine connections need if not modification then serious reconsideration. It should be noted that there were a few opinions regarding the generalization of Dirac spinor in general relativity proposed by Fock [40–42], Pauli [43], Sommerfeld [44], Wigner [45] and others. We plan to address this issue in near future.

**Funding:** This research received no external funding.

**Institutional Review Board Statement:** Not applicable.

**Informed Consent Statement:** Not applicable.

**Data Availability Statement:** No datasets were generated or analysed during the current study.

**Acknowledgments:** This paper has been supported by the RUDN University Strategic Academic Leadership Program.

**Conflicts of Interest:** The author declares no conflict of interest.

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
