# Peer review of "Spinor Field in FLRW Cosmology"

_universe, doi:10.3390/universe9050243_

Round 1

Reviewer 1 Report

In this paper, the author reviews the role played by the non-linear spinor field in the evolution of the universe in FLRW cosmological model. In particular, the author revises the results obtained previously in Cartesian and spherical coordinates and discusses some interesting applications.

The author discusses the dynamics of the spinor field, both massive and massless, and its energy-momentum tensor in FLRW with a detailed analysis of the coordinate dependence energy-momentum components. Next, the author presents a numerical analysis of the barotropic equation of states, Chaplygin gas, modified Chaplygin gas and modified quintessence models. 

All in all, the paper contains an interesting review and a clear analysis that can be useful to a wide range of audience. I would only suggest the correction of the misprints from the eq. (6) and a minor revision of English.

I recommend the paper for publication.

I recommend a minor revision of English.

Reviewer 2 Report

The author argues that in the FRLW cosmological model with a non-minimally coupled nonlinear spinor field as a matter field, a coordinate transformation from Cartesian coordinates to spherical coordinates generates changes in the energy-momentum tensor of the spinor field that have physical consequences. However, general relativity can be given in a coordinate-free formalism that don't contain coordinate transformations. I would like to learn why this is not the case in the theory under consideration. Perhaps the spinor affine connection can be modified so that the energy-momentum tensor don't depend nontrivially on the coordinate system.

Reviewer 3 Report

The refereed paper is devoted to the study of the behaviour of the non-linear spinor field in the open, closed and flat Friedmann universes and to its possible role as a driving force of the cosmological evolution. The paper contains some interesting technical details, which can be useful for the researchers working in this field and I think that this paper can be published in the Universe.
